# Beyond Disentanglement: On the Orthogonality of Learned Representations

## Abstract

Evaluating learned representations independently of designated downstream tasks is pivotal for crafting robust and adaptable algorithms across a diverse array of applications. Among such evaluations, the assessment of disentanglement in a learned representation has emerged as a significant technique. In a disentangled representation, independent data generating factors are encoded in mutually orthogonal subspaces, a characteristic enhancing numerous downstream applications, potentially bolstering interpretability, fairness, and robustness. However, a representation is often deemed well-disentangled if these orthogonal subspaces are one-dimensional and align with the canonical basis of the latent space – a powerful yet frequently challenging or unattainable condition in real-world scenarios – thus narrowing the applicability of disentanglement. Addressing this, we propose a novel evaluation scheme, Importance-Weighted Orthogonality (IWO), to gauge the mutual orthogonality between subspaces encoding the data generating factors, irrespective of their dimensionality or alignment with the canonical basis. For that matter, we introduce a new method, Latent Orthogonal Analysis (LOA), which identifies the subspace encoding each data generating factor and establishes an importance-ranked basis spanning it, thereby forming the foundational bedrock for IWO. Through extensive comparisons of learned representations from synthetic and real-world datasets, we demonstrate that, relative to existing disentanglement metrics, IWO offers a superior assessment of orthogonality and exhibits stronger correlation with downstream task performance across a spectrum of applications.

## 1 Introduction

Learning meaningful representations, independent of specific downstream tasks, is central to representation learning. Such representations should ideally capture human-centric inductive biases, and much of the research emphasis revolves around determining how various models, training methodologies, or hyperparameters achieve this goal. The aim is not just to advance model development but also to pave the way for algorithms that are robust, adaptable, and efficient across a multitude of applications.

Among various such methods, disentangled representation learning emerged as a prominent technique. In this framework, data $x$ is often assumed to be generated by an underlying function $g$ driven by ground truth, generative factors $\{z_j\}_{j=1}^{K}$ and other variability factors $\psi$, that is $x = g(z, \psi)$. A model then learns a mapping $c = r(x) \in \mathbb{R}^L$ from the data to a latent representation space. A common characterization of disentanglement posits that the generative factors are represented by single distinct components of $c$, implying that they manifest as orthogonal 1-d latent subspaces aligned with the canonical basis within the latent representation, up to a scaling factor and irrelevant latent dimensions (*i.e.*, when $L > K$). Such properties in learned representations have been pivotal for the development of models that are: (i) interpretable, by disentangling the latent factors, individual variables can be analyzed, making the models more comprehensible and transparent (Zhu et al. (2021); Klein et al. (2022) (ii) flexible, as disentangled representations result from models that can be adapted to a range of tasks without extensive retraining, (iii) fair, by identifying and isolating sensitive features in the representation, it is possible to build models that are less prone to biased predictions (Creager et al. (2019)). We argue that among the properties that a disentangled representation possesses, the orthogonality of independent human-centric concepts and categories in the

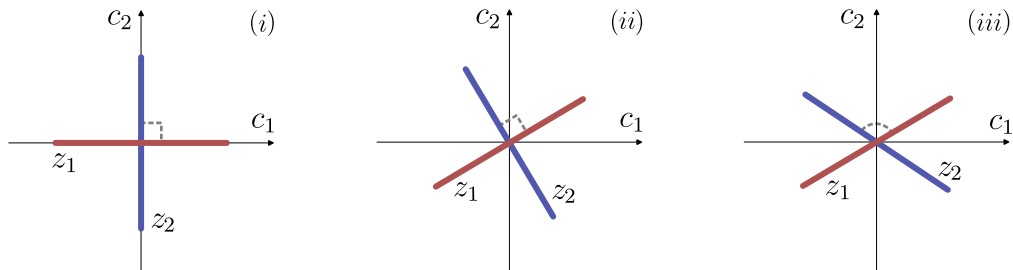

Figure 1: Three configurations of a 2-dimensional latent space. The red and blue lines represent the latent subspaces where the generative factors $z_1$, $z_2$ lie. (i) The generative factors are disentangled since they are singularly coincident with a single component in the latent representation. This corresponds to perfect scores for both the DCI-D and the IWO metric. (ii) the generative factors are not disentangled, however they preserve the orthogonality. In this case, DCI-D is low, while IWO stays optimal. (iii) the generative factors are neither disentangled nor orthogonal, both DCI-D and the IWO metric are low.

latent space plays a fundamental role in why disentangled representations can boost downstream task performance. Consider for example a latent space encoding human faces, where the concept of smiling lies in a potentially multidimensional subspace orthogonal to other facial feature concepts. Using such a latent representation, a downstream model tasked to convert non-smiling faces to smiling faces might only require a few annotated examples. A similar argument holds for the embedding of natural language. If the concept of age lies in a dimension orthogonal to most other concepts, converting baby to adult or puppy to dog becomes as easy as identifying such dimension and navigating through it.

However, even if a model learns perfectly orthogonal subspaces for independent generative factors, it is not considered disentangled *per se*. When orthogonal subspaces are not (i) 1-dimensional, (ii) aligned with the canonical basis of the representation, (iii) showing a linear behavior with respect to their related generative factors, the latent space might, in fact, be considered very badly disentangled according to popular metrics such as MIG (Chen et al., 2018b) or DCI-D (Eastwood & Williams, 2018). Disentanglement then becomes a restrictive concept for the evaluation with respect to many downstream applications, which benefit from the orthogonality of human-centric concepts. In Figure 1 we present three cases to highlight the differences between disentangled and orthogonal representations, in terms of DCI-D and IWO scores.

Motivated by these considerations, we propose a new metric, *Importance-Weighted Orthogonality (IWO)*, which addresses these discrepancies by assessing the orthogonality of generative factor subspaces based on the magnitude of their projections onto each other. In essence, our metric generalizes the notion of cosine similarity to a multidimensional context. IWO relies on Latent Orthogonal Analysis (LOA), a novel methodology that identifies the "latent subspaces" where each generative factor actually varies, while accommodating any potential non-linear dynamics within such subspaces. Each dimension of the latent subspace is then assigned an "importance score" that quantifies the variability of the generative factor along that dimension. Subsequently, we compute the mutual projections between all pairs of identified subspaces and aggregate them to derive a final score which estimates the orthogonality of the latent representation.

We conduct extensive experimental analysis on synthetic datasets showing that our metric outperforms other metrics in measuring the orthogonality of subspaces. Additionally, we show that our metric is equally or better suited for model selection for a range of downstream applications compared to other metrics commonly used in the literature.

## 2 RELATED WORK

### 2.1 DISENTANGLEMENT METRICS

Alongside the task of disentanglement, gauging a model's performance in disentangling a representation has emerged as a non-trivial problem. Beyond visual inspection of the results, a variety of

quantitative methodologies have been developed to tackle this issue. Higgins et al. (2017) proposed to measure the accuracy of a classifier predicting the position of a fixed generative factor. Kim & Mnih (2018a) further robustify the metric by proposing a majority voting scheme related to the least-variance factors in the representations. Chen et al. (2018b) introduce the *Mutual Information Gap* estimating the normalized difference of the mutual information between the two highest factors of the representation vector. Eastwood & Williams (2018) propose the DCI metrics to evaluate the correlation between the representation and the generative factors. For each of them, one linear regressor is trained and the entropy over the rows (*Disentanglement*) and the columns (*Completeness*) is computed, along with the error (*Informativeness*) achieved by each regressor. The *Modularity* metric introduced by Ridgeway & Mozer (2018) computes the mutual information of each component of the representation to estimate its dependency with at most one factor of variation. *SAP* score (Kumar et al., 2018) estimates the difference, on average, of the two most predictive latent components for each factor.

The use of metrics such as the aforementioned ones contributed to shaping several definitions of disentanglement, each encoding a somewhat different aspect of disentangled representations, which led to a fragmentation of definitions (*cf.* Locatello et al. (2019)). Higgins et al. (2018) attempted instead to propose a unified view of the disentanglement problem, by defining *Symmetry-Based Disentangled Representation Learning (SBDRL)*, a principled framework drawn from group representation theory. The authors established disentanglement in terms of a morphism from world states to decomposable latent representations, equivariant with respect to decomposable symmetries acting on the states/representations. For a representation to be disentangled, each symmetry group must act only on a corresponding (multidimensional) subspace of the representation. Following this conceptualization, Caselles-Dupré et al. (2019) demonstrated the learnability of such representations, provided the actions and the transitions between the states. Painter et al. (2020) extended the work by proposing a reinforcement learning pipeline to learn without the need for supervision. Note worthy are the two proposed metrics: (i) an *independence score* that, similarly to our work, estimates the orthogonality between the generative factors in the fashion of a canonical correlation analysis; (ii) a *factor leakage score*, extended from the MIG metric to account for all the factors. Tonnaer et al. (2022) formalized the evaluation in the SBDRL setting and proposed a principled metric that quantifies the disentanglement by estimating and applying the inverse group elements to retrieve an untransformed reference representation. A dispersion measure of such representations is then computed. Note that while most works focus on the linear manipulation of the latent subspace, the SBDRL can be also used in non-linear cases. Differently from our framework, however, SBDRL requires modelling the symmetries and the group actions, which may be challenging in scenarios where there is not a clear underlying group structure (Tonnaer et al., 2022).

Recently, several works such as those of Montero et al. (2021), Träuble et al. (2021) and Dittadi et al. (2021) have proposed to go beyond the notion of disentanglement, advocating for the relaxation of the independence assumption among generative factors – perceived as too restrictive for real-world data problems – and modelling their correlations. Reddy et al. (2022) and Suter et al. (2019) formalized the concept of causal factor dependence, where the generative factors can be thought of as independent or subject to confounding factors. The latter work introduced the *Interventional Robustness Score* assessing the effects in the learned latent space when varying its related factors. Valenti & Bacciu (2022) defined the notion of *weak disentangled* representation that leaves correlated generative factors entangled and maps such combinations in different regions of the latent space.

## 2.2 EXPLICITNESS

In this paper, we aim to move beyond the definition of disentanglement and characterize less restrictive, nonetheless fundamental, properties of the latent representations (*i.e.*, mutual orthogonality among the latent subspaces). The authors of Eastwood et al. (2023) also relaxed the notion of disentanglement, by extending the DCI metric with an Explicitness (E) metric:

$$E(z_j, \boldsymbol{c}; \mathcal{F}) = 1 - \frac{\text{AULCC}(z_j, \boldsymbol{c}; \mathcal{F})}{Z(z_j; \mathcal{F})}, \tag{1}$$

where $z_j$ and $\boldsymbol{c}$ represent a generative factor and the latent space respectively, and $\mathcal{F}$ a class of regressors (*e.g.*, multilayer perceptrons or random forests). AULCC is the Area Under the Loss Curve, computed by recording the minimum losses achievable by regressing $z_j$ from $\boldsymbol{c}$ with models

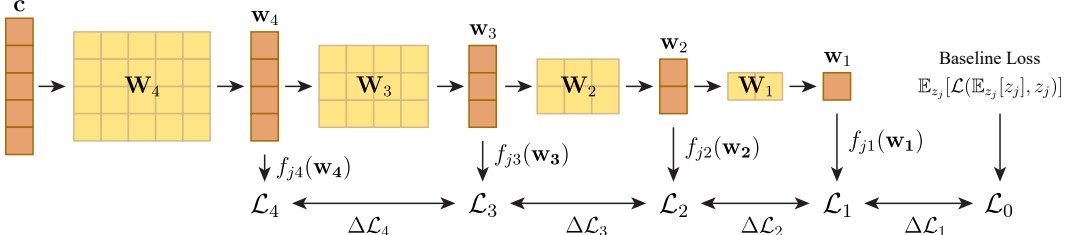

Figure 2: Model architecture and training paradigm for identifying subspaces of maximal importance and assessing their respective significance for generative factor $z_j$. Through iterative multiplications with $\boldsymbol{W}_l \in \mathbb{R}^{l \times l+1}$, the input is projected to subspaces of decreasing dimensions. The resulting outputs $\boldsymbol{w}_d$ are directed into NN heads, denoted as $f_{j,l} : \mathbb{R}^l \rightarrow \mathbb{R}$. The importance is gauged by the loss decrease $\Delta\mathcal{L}_l$ between consecutive NNs $f_{j,l}, f_{j,l-1}$. This training facilitates the optimization on lower-dimensional projections by steering them towards their optimal subspaces, ensuring that smaller subspaces are nested within the larger ones.

in $\mathcal{F}$ of increasing capacity. The denominator, easily computable, acts as a normalizing constant, so that $E \in [0, 1]$. As an example, $E = 1$ suggests that a linear regressor is sufficient to reach zero error, proving that the representation is efficient. To account for a bias toward large representations, the explicitness is paired with the Size (S), computed as the ratio between the number of generative factors and the size of the latent representation.

The DCI-ES framework tries to quantify a fairly general aspect of the representation, the explicitness, related to the capacity required to regress the representation to its generative factors. Our metric instead measures the mutual orthogonality between subspaces associated to generative factors, filtering the relationship between the generative factors and their latent subspaces. To clarify the difference, consider the four different latent spaces described in Figure 4 (*cf.* Appendix).

## 3 METHODOLOGY

In this section, we first present Latent Orthogonal Analysis (LOA), a technique for identifying the latent subspaces where each generative factor lies. We then turn to Importance-Weighted Orthogonality (IWO), which estimates the mutual orthogonality of these subspaces. Additionally we also present Importance Weighted Rank (IWR), which measures the rank of subspaces found with LOA.

### 3.1 LATENT ORTHOGONAL ANALYSIS (LOA)

Consider a latent representation or *code*, $\boldsymbol{c} \in \mathbb{R}^L$ encoding the generative factors $(z_1, \ldots, z_K) \in \mathbb{R}^K$ with $L \geq K$. Note that $z_j = f_j(\boldsymbol{c})$, with $f_j$ being a potentially complex non-linear function. However, not all changes in $\boldsymbol{c}$ imply a change in $z_j$. In particular, we define the *invariant latent subspace* of $z_j$ to be the largest linear subspace $\mathbb{I}_j \subseteq \mathbb{R}^L$, such that $f_j(\boldsymbol{c} + \boldsymbol{v}) = f_j(\boldsymbol{c})$, $\forall \boldsymbol{v} \in \mathbb{I}_j$. Accordingly, the *variant latent subspace* (simply *latent subspace*) of $z_j$ is defined to be the orthogonal complement of $\mathbb{I}_j$ and will be denoted as $\mathbb{S}_j$, with dimensionality $R_j$.

So far, our definition is entirely agnostic to the "importance" of dimensions spanning the latent subspace. However, it is likely that a generative factor exhibits different variability along the dimensions spanning its subspace. In a manner akin to Linear Principal Component Regression, our objective is to determine an orthonormal basis that spans the latent subspace $\mathbb{S}_j$. The vectors in this basis are organized based on their relative importance for predicting the factor $z_j$. We refer to this specialized basis as importance ordered orthonormal (i.o.o.). In the next paragraphs, we describe how to find such a basis.

**Subspace learning** We propose an iterative procedure which, starting from a code $\boldsymbol{c} \in \mathbb{R}^L$, projects it onto progressively smaller dimensional subspaces, removing the least important dimension for regressing $z_j$ at each step, until the subspace is 1-dimensional. In particular, we design a Linear Neural Network (LNN), composed of a set of projective transformations $\boldsymbol{W}_L, \ldots, \boldsymbol{W}_1$ which reduce the dimensionality of $\boldsymbol{c}$ step-by-step. No non-linearities are applied, therefore each

layer performs a projection onto a smaller linear subspace. We assume a reduction factor of 1, so that $\boldsymbol{W}_l \in \mathbb{R}^{l \times l+1}$ for $l = 1, \ldots, L$, however higher values can also be considered, especially when $L$ is large.

At each layer of the LNN, the intermediate projection $\boldsymbol{w}_l \in \mathbb{R}^l$ is passed as input to a non-linear neural network $f_{jl}$ regressing $z_j$. Weights are not shared between the regressors $f_{j1}, \ldots, f_{jL}$. As such, they operate on linear subspaces of the original latent space, revealing any non-linear manifestation of the generative factor. Training of all neural networks is performed in parallel. In principle, the gradient flow from each individual regressor $f_{jl}$ can be stopped after the corresponding $\boldsymbol{W}_l$, however, we attested a faster convergence when letting the gradient of each $f_{jl}$ flow back up to $\boldsymbol{c}$. The purpose of the parallel exploration of all nested subspaces is to ease the search for the smallest subspaces by finding the larger ones first.

When training has ended, each regressor $f_{jl}$ can be associated with an expected loss of regressing the factor. Let us denote this expected loss as $\mathcal{L}_l = \mathbb{E}_{\boldsymbol{c}} \left[ \ell(f_{jl}(\boldsymbol{w}_l(\boldsymbol{c})), z_j(\boldsymbol{c})) \right]$, where $\ell$ is a specific loss term. In particular, note that $\mathcal{L}_{l-1} \leq \mathcal{L}_l$ because of the potential information loss due to dimensionality reduction. Let us now quantify the loss-increase by each of the LNN-projections as $\Delta \mathcal{L}_l = \mathcal{L}_{l-1} - \mathcal{L}_l$. We define $R_j$ as the smallest dimensionality at which the lowest achievable loss is reached, that is, $\Delta \mathcal{L}_l = 0$ for $l > R_j$. For $l = 1$, we compute $\Delta \mathcal{L}_1$ as the difference between $\mathcal{L}_1$ and a baseline loss $\mathcal{L}_0 = \mathbb{E}_{z_j} \left[ \ell(\mathbb{E}_{z_j} [z_j], z_j) \right]$. The entire learning process is depicted in Figure 2.

**Basis generation** Using the trained weights $\boldsymbol{W}_L, \ldots, \boldsymbol{W}_1$ of the LNN, along with the layer-specific loss differences $\Delta \mathcal{L}_l$, we now describe how to construct an i.o.o. basis, $\{\boldsymbol{b}_l \in \mathbb{R}^L \mid l = 1, \ldots, R_j\}$, spanning factor $z_j$'s latent subspace. Note that each layer $\boldsymbol{W}_l$ of the LNN effectively eliminates a single dimension from the data representation. The training methodology ensures that the dimension removed at each layer is the least important one for the regression of $z_j$, as determined by causing the smallest increase in $\Delta \mathcal{L}_{l+1}$. Therefore, a forward pass through the entire LNN effectively projects any input vector $c$ onto the most important dimension. We define $\boldsymbol{b}_1$ to span this dimension in the original representation space $\mathbb{R}^L$. Together with the dimensions sequentially removed between $\boldsymbol{W}_1$ and $\boldsymbol{W}_{R_j}$ we can thus form an orthogonal basis for $\mathbb{S}_j$, in decreasing order of importance. To retrieve the dimension removed by each layer $\boldsymbol{W}_l$ in the form of a basis vector $\boldsymbol{b}_{l+1}$, we first perform reduced QR decomposition for all weight matrices $\boldsymbol{W}_l$. For each resulting $\boldsymbol{Q}_l \in \mathbb{R}^{l \times (l+1)}$ we then define $\boldsymbol{q}_l^\perp \in \mathbb{R}^{l+1}$, as the normalized *row* vector perpendicular to all rows in $\boldsymbol{Q}_l$. The basis vector $\boldsymbol{b}_l$ can then be calculated as

$$\boldsymbol{b}_l^\top = \boldsymbol{q}_{l-1}^\perp \boldsymbol{Q}_l \prod_{d=l+1}^L \boldsymbol{W}_d \quad \text{for } l = 1, \ldots, R_j, \tag{2}$$

with $\boldsymbol{q}_0 = 1$, We can quantify the importance $\mathcal{I}_l$ of each basis vector $\boldsymbol{b}_l$ by the relative loss increase associated to its layer in the LNN:

$$\mathcal{I}_l = \frac{\Delta \mathcal{L}_l}{\mathcal{L}_0 - \mathcal{L}_{R_j}} \quad \text{for } l = 1, \ldots, R_j. \tag{3}$$

Finally, we normalize each vector $\boldsymbol{b}_l$ obtaining an i.o.o. basis for $\mathbb{S}_j$.

## 3.2 ORTHOGONALITY OF SUBSPACES

For the derivation of the IWO metric, consider two matrices $\boldsymbol{B}_j \in \mathbb{R}^{R_j \times L}$, $\boldsymbol{B}_k \in \mathbb{R}^{R_k \times L}$ whose rows compose the i.o.o. basis vectors spanning $z_j$'s and $z_k$'s latent subspaces respectively. We define the orthogonality between the two latent subspaces as

$$\mathrm{O}(\mathbb{S}_j, \mathbb{S}_k) = \frac{\mathrm{Tr}(\boldsymbol{B}_j \boldsymbol{B}_k^\top \boldsymbol{B}_k \boldsymbol{B}_j^\top)}{\min(R_j, R_k)}. \tag{4}$$

Note that the trace $\mathrm{Tr}(\boldsymbol{B}_j \boldsymbol{B}_k^\top \boldsymbol{B}_k \boldsymbol{B}_j^\top)$ equals the sum of the squared values in $\boldsymbol{B}_j \boldsymbol{B}_k^\top$, *i.e.*, $\sum_{l,m} (\boldsymbol{B}_j \boldsymbol{B}_k^\top)_{ml}^2 = \sum_{l,m} (\boldsymbol{b}_{jl} \cdot \boldsymbol{b}_{km})^2$, where $\boldsymbol{b}_{jl}$ and $\boldsymbol{b}_{km}$ are the $l$-th and $m$-th rows of $\boldsymbol{B}_j$ and $\boldsymbol{B}_k$ respectively. The maximum of the trace is therefore $\min(R_j, R_k)$, reached if $\mathbb{S}_j$ is a subspace of $\mathbb{S}_k$ or viceversa. It therefore holds that $\max \mathrm{O}(\mathbb{S}_j, \mathbb{S}_k) = 1$. The minimum of $\mathrm{O}(\mathbb{S}_j, \mathbb{S}_k)$ is 0, which is the case when $\mathbb{S}_k$ lies in the invariant latent subspace of $z_j$. This definition of orthogonality can be interpreted as the average absolute cosine similarity between any vector pair from $\mathbb{S}_j$ and $\mathbb{S}_k$.

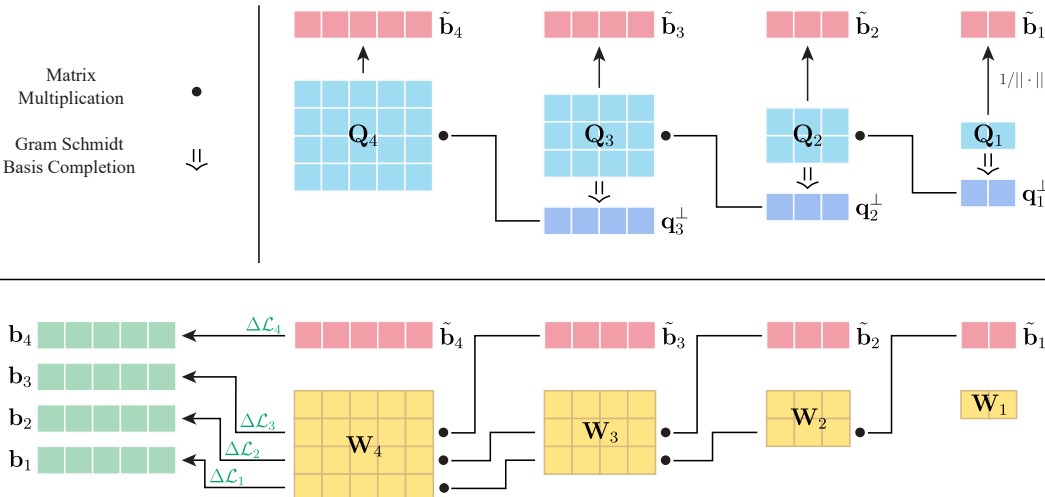

Figure 3: **Dimension Identification (top)**: Matrices $\boldsymbol{Q}_l$ are obtained by $QR$-decomposition on the learned weights $\boldsymbol{W}_l$. We allocate the projected basis vectors $\boldsymbol{b}_l$ by finding $\boldsymbol{q}_{l-1}$ perpendicular to all rows in $\boldsymbol{Q}_{l-1}$ and multiplying it with $\boldsymbol{Q}_l$. **Basis generation (bottom)**: Basis vector projections $\tilde{\boldsymbol{b}}_{l+1}$ are iteratively multiplied with $\boldsymbol{W}_l$. The resulting vectors are normalized to form an orthonormal basis. The importances correspond to the (normalized) loss decreases found in the learning phase.

### 3.3 IMPORTANCE WEIGHTED ORTHOGONALITY (IWO)

Based on this definition, we construct IWO. As the name suggests, in addition to encapsulating the orthogonality between factor subspaces, IWO also takes into consideration the importance of the dimensions spanning them.

To calculate the importance-weighted projection of $z_j$'s subspace onto $z_k$'s subspace, we first scale the corresponding bases vectors in $\boldsymbol{B}_j, \boldsymbol{B}_k$ with their respective importance before projecting them onto one another. IWO is the sum of all individual projections. Using $\boldsymbol{U}_j = \boldsymbol{D}_j \boldsymbol{B}_j$, where $\boldsymbol{D}_j \in \mathbb{R}^{R_j \times R_j}$ is diagonal with the $l$-th diagonal entry corresponding to the square root of importance, $\sqrt{\mathcal{I}_l(z_j)}$, we can efficiently calculate IWO as:

$$\text{IWO}(z_j, z_k) = \text{Tr}(\boldsymbol{U}_j \boldsymbol{U}_k^\top \boldsymbol{U}_k \boldsymbol{U}_j^\top) \tag{5}$$

Similar to the orthogonality, $\text{IWO}(z_j, z_k) \in [0, 1]$. However, $1$ is only reached if $z_j$ and $z_k$, in addition to lying in the same subspace, also share the same importance along the same dimensions. Together with IWO, and in analogy to the *Completeness* metric of the DCI framework, we define an importance-weighted rank (IWR) for each generative factor:

$$\text{IWR}(z_j) = 1 - \mathcal{H}_j', \tag{6}$$

where $\mathcal{H}_j' = -\sum_{l=1}^{R_j} \mathcal{I}_l(z_j) \log_{R_j} \mathcal{I}_l(z_j)$. IWR thus measures how the importance is distributed among the $L$ dimensions of the subspace. Note that, for a particular $R_j$, $\text{IWR}(z_j)$ is minimized if the importance is distributed equally along all $R_j$ dimensions spanning $z_j$'s subspace. In that case, $\text{IWR}(z_j) = 0$. We denote the mean over all generative factors of IWO and IWR as $\overline{\text{IWO}}$ and $\overline{\text{IWR}}$.

LOA allocates an i.o.o. basis for each generative factor of a learned representation. However, when comparing representations, we also have to account for differences in $\mathcal{L}_{R_j}$, as this loss corresponds to the best possible regression of $z_j$ from the representation. In the DCI framework, this aspect is captured by the Informativeness metric. In order not to favour representations with low $R_j$ and high $\mathcal{L}_{R_j}$ over those with low $\mathcal{L}_{R_j}$ and higher $R_j$, we adjust the importance weights of any factor $z_j$ whose $\mathcal{L}_{R_j} > 0$. To do that, we first complete the factor's basis $\boldsymbol{B}_j$ to span the whole latent space, then we distribute the loss $\mathcal{L}_{R_j}$ among the importance of the basis vectors equally:

$$\mathcal{I}_l = \frac{\Delta \mathcal{L}_l + \mathcal{L}_{R_j}/L}{\mathcal{L}_0} \quad \text{for } l = 1, \dots, L. \tag{7}$$

Lastly, we adjust $R_j = L$.

## 4 EXPERIMENTS

To test the effectiveness of our IWO and IWR implementations, we first test whether we can (1) recover the true latent subspaces and (2) correctly assess their IWO and IWR. For that purpose, we set up a synthetic data generation scheme, providing us with the ground truth IWO and IWR values. For comparison purposes, we also test how other metrics assess the synthetic data, namely *Disentanglement*, *Completeness*, *Informativeness* and *Explicitness*, as measured by the DCI-ES framework.

Finally, we evaluate our IWO implementation on three disentanglement datasets, dSprites (Matthey et al., 2017), Cars3D (Reed et al., 2015) and SmallNorbs LeCun et al. (2004). We measure how strong IWO and the commonly used DCI-D and MIG metrics correlate with downstream task performance for several common variational autoencoder models, over a wide range of seeds and hyperparameters. More details are listed in the appendix and in our open-source code implementation[1].

### 4.1 SYNTHETIC EXPERIMENTS

We introduce a synthetic data generating scheme, which generates vectors of i.i.d Gaussian distributed latent representations $c \in \mathbb{R}^L$. Then, on the basis of the latent representations, we synthesize $K$ generative factors $\{z_1, \ldots, z_K\}$. For simplicity, we choose $L$ as a multiple of $K$.

For simulating a disentangled latent space, we define each $z_j$ to be linearly dependent on a single, distinct element of $c$. In order to assess higher dimensional cases with non-linear relationships, we consider a non-linear commutative mapping $f : \mathbb{R}^{R_j} \to \mathbb{R}$. In particular, we experiment with a polynomial (Poly.) and a trigonometric (Trig.) $f$. Notice that the commutativity enforces that the distribution is spread evenly across all dimensions, such that we can easily assess the performance of IWR($z_j$). For simplicity, we always set $R_j = R$ for all $j = 1, \ldots, K$. Together, $L$ (latent space dimension), $K$ (number of factors) and $R$ (latent subspace dimension) determine how many dimensions each generative factor shares with the others. We consider the shared dimensions to be contiguous. To give an intuition, consider the following two examples:

- $L = 10$, $K = 5$, $R = 2$. Each $z_j$ is a function of two elements of $c$, namely $z_1 = f(c_1, c_2)$, $z_2 = f(c_3, c_4), \ldots, z_5 = f(c_9, c_{10})$. No dimensions are shared, thus $\overline{\text{IWO}} = 0$, $\overline{\text{IWR}} = 2$.
- $L = 10$, $K = 5$, $R = 5$. Each $z_j$ is a function of five successive elements of $c$, namely $z_1 = f(c_1, c_2, c_3, c_4, c_5)$, $z_2 = f(c_3, c_4, c_5, c_6, c_7), \ldots, z_5 = f(c_9, c_{10}, c_1, c_2, c_3)$. On average, each $z_j$ shares two of its five dimensions, thus $\overline{\text{IWO}} = 0.2$, $\overline{\text{IWR}} = 5$.

To test representations that are not aligned with the canonical basis, we apply Random Orthogonal Projections (ROP) $R \in \mathbb{R}^{L \times L}$ to $c$. In line with commonly used datasets such as Cars3D or dSprites, we rescale and quantize the values in the $z_j$ to the range $[0, 1]$.

All experiments are run four times with differing random seeds. The standard deviation was smaller than 0.02 for all reported values. The results are displayed in Table 1.

In the first experiment (Exp. 1), we define two setups, both with $L = K = 5$ and $R = 1$. First, we let $z$ be a mere permutation of $c$, second we let $z$ be equal to $c + \epsilon$, with $\epsilon \sim \mathcal{N}(0, 0.01)$ being Gaussian noise. The results display proficient disentanglement assessment by $DCI$, and near-perfect recovery in $\overline{\text{IWO}}$ and $\overline{\text{IWR}}$. However, Explicitness deviates from the expected value of 1.

In the second experiment (Exp. 2), we choose $L = 10$, $K = 5$ and $R = 2$. No dimensions are shared between the generative factors. The mapping $f$ is polynomial. We also apply ROP in one of the experiments. We see that, as expected, applying an ROP dramatically lowers both $D$ and $C$, while $E$, $\overline{\text{IWO}}$ and $\overline{\text{IWR}}$ are resilient to it.

In the third experiment (Exp. 3), we set $R_j = 5$. Each generative factor now shares, on average, two out of five dimensions with the others. We observe that while $D$, $C$, $\overline{\text{IWO}}$ and $\overline{\text{IWR}}$ are sensitive to the shared dimensions, $E$ is almost unchanged with respect to our second experiment. Again, $\overline{\text{IWO}}$ and $\overline{\text{IWR}}$ are recovered almost perfectly.

In the fourth experiment (Exp. 4), we set $L = 20$ and vary values for $R$, ROP and the encoding function. We observe that neither $D$, $C$ nor $E$ can be used to assess the orthogonality of the latent

---

[1] https://anonymous.4open.science/r/iwo-E0C6/README.md

Table 1: Comparison of (D) Disentanglement, (C) Completeness, (I) Informativeness, (E) Explicitness, $\overline{\text{IWO}}$ and $\overline{\text{IWR}}$ for Polynomial (Poly.) and Trigonometric (Trig.) encodings. ROP denotes a random orthogonal projection.

| | $L$ | $K$ | $R$ | Func. | ROP | $D\uparrow$ | $C\uparrow$ | $I\uparrow$ | $E\uparrow$ | $\overline{\text{IWO}}\downarrow$ | $\overline{\text{IWR}}\uparrow$ |
|---|---|---|---|---|---|---|---|---|---|---|---|
| Exp. 1 | 5 | 5 | 1 | Noisy | No | 0.98 | 0.98 | 1.00 | 0.90 | 0.01 | 1.00 |
| | 5 | 5 | 1 | Perm. | No | 0.98 | 0.98 | 1.00 | 0.90 | 0.01 | 1.00 |
| Exp. 2 | 10 | 5 | 2 | Poly. | No | 0.99 | 0.69 | 1.00 | 0.75 | 0.01 | 0.70 |
| | 10 | 5 | 2 | Poly. | Yes | 0.21 | 0.15 | 1.00 | 0.75 | 0.01 | 0.70 |
| Exp. 3 | 10 | 5 | 5 | Poly. | No | 0.41 | 0.30 | 1.00 | 0.74 | 0.40 | 0.31 |
| | 10 | 5 | 5 | Poly. | Yes | 0.06 | 0.04 | 1.00 | 0.74 | 0.40 | 0.31 |
| | 10 | 5 | 5 | Trig. | No | 0.41 | 0.30 | 0.99 | 0.73 | 0.41 | 0.30 |
| | 10 | 5 | 5 | Trig. | Yes | 0.06 | 0.04 | 1.00 | 0.73 | 0.40 | 0.31 |
| Exp. 4 | 20 | 5 | 4 | Poly. | No | 0.98 | 0.53 | 0.99 | 0.76 | 0.01 | 0.54 |
| | 20 | 5 | 4 | Poly. | Yes | 0.12 | 0.07 | 1.00 | 0.76 | 0.01 | 0.54 |
| | 20 | 5 | 8 | Poly. | No | 0.56 | 0.30 | 0.99 | 0.74 | 0.25 | 0.31 |
| | 20 | 5 | 8 | Poly. | Yes | 0.05 | 0.03 | 0.99 | 0.74 | 0.25 | 0.31 |
| | 20 | 5 | 4 | Trig. | No | 0.96 | 0.52 | 0.99 | 0.73 | 0.00 | 0.53 |
| | 20 | 5 | 4 | Trig. | Yes | 0.12 | 0.07 | 0.99 | 0.74 | 0.00 | 0.53 |
| | 20 | 5 | 8 | Trig. | No | 0.54 | 0.29 | 0.98 | 0.71 | 0.25 | 0.31 |
| | 20 | 5 | 8 | Trig. | Yes | 0.07 | 0.04 | 0.98 | 0.70 | 0.25 | 0.30 |

representation reliably. While $E$ does slightly decrease for increased $R$, its variability due to different complexities of the encoding function overshadows this. Meanwhile, $\overline{\text{IWO}}$ and $\overline{\text{IWR}}$ are able to reliably assess both the orthogonality and the rank of the representation almost perfectly.

### 4.2 Downstream Experiments

We also assess $\overline{\text{IWO}}$'s and $\overline{\text{IWR}}$'s correlation with downstream task performance along with the DCI-D, DCI-C and MIG metrics, computed on three common benchmarks of the *disentanglement lib* framework Locatello et al. (2019). In particular, we consider DSprites (Matthey et al., 2017), Cars3d (Reed et al., 2015) and smallNORB LeCun et al. (2004). We test four standard models, namely $\beta$-VAE Higgins et al. (2016), Annealed VAE (Burgess et al., 2018), $\beta$-TCVAE Chen et al. (2018a) and Factor-VAE Kim & Mnih (2018b). For each model, we consider six different regularization strengths each with ten different random seeds. For each dataset, we consider two downstream tasks, namely regressing the generative factors via logistic regression (T1), and regressing the generative factors via boosted trees (T2). Correlations are calculated between tasks and averaged metrics, where the average is taken over the seeds and grouped by hyperparameter. As such, we assess the capability of the metrics to point us to good hyperparameters.

For DSprites, the correlation with downstream tasks is high for all metrics and tasks. IWR, DCI-D, DCI-C and MIG exhibit a slightly higher correlation with (T2) than IWO for most models. This changes for (T1), where IWO exhibits a slightly higher correlation than other metrics for most models. However, no metric emerges as definitely superior, as all correlation factors are quite high.

For Cars3D, DCI-D, DCI-C and MIG fail to correlate with downstream tasks. This was also noted by Locatello et al. (2019). Instead, IWO and IWR fairly correlate for both tasks and most models.

For the SmallNorb dataset, DCI-D, DCI-C and MIG fairly correlate for most models on both tasks. However, again, IWO and IWR seem to correlate more constantly over the models and tasks, not exhibiting negative correlation outliers.

## 5 Discussion and Conclusion

**Summary of Key Findings** In our investigation, we pivoted from the conventional focus on the disentanglement of generative factors to a novel examination of their orthogonality. Our approach,

Table 2: Correlation coefficients between IWO, IWR, DCI-D, DCI-C, MIG and the downstream task of recovering the generative factors with Logistic Regression (T1) and Boosted Trees (T2). Examined Latent representations of DSprites, Cars3D and small Norb datasets learned by models: Annealed VAE (A-VAE), $\beta$-VAE, $\beta$-TCVAE and Factor-VAE (F-VAE).

| | Model | 1 - $\overline{\text{IWO}}$ | | $\overline{\text{IWR}}$ | | D | | C | | MIG | |
|---|---|---|---|---|---|---|---|---|---|---|---|
| | | T1 | T2 | T1 | T2 | T1 | T2 | T1 | T2 | T1 | T2 |
| **DSprites** | A-VAE | 0.87 | 0.93 | 0.88 | 0.93 | 0.84 | 0.99 | 0.82 | 0.99 | 0.85 | 0.97 |
| | $\beta$-VAE | 0.02 | 0.70 | 0.06 | 0.98 | 0.03 | 0.98 | 0.07 | 0.99 | 0.01 | 0.97 |
| | $\beta$-TCVAE | 0.52 | 0.96 | 0.45 | 0.99 | 0.37 | 1.00 | 0.38 | 0.99 | 0.25 | 0.99 |
| | F-VAE | 0.30 | 0.94 | 0.27 | 0.94 | 0.43 | 0.96 | 0.48 | 0.94 | 0.35 | 0.96 |
| **Cars3D** | A-VAE | 0.82 | 0.76 | 0.88 | 0.68 | -0.40 | 0.72 | -0.85 | 0.18 | -0.82 | 0.19 |
| | $\beta$-VAE | 0.79 | 0.71 | 0.95 | 0.69 | -0.93 | -0.92 | -0.95 | -0.94 | -0.92 | -0.93 |
| | $\beta$-TCVAE | 0.83 | 0.80 | -0.42 | -0.12 | -0.94 | -0.92 | -0.97 | -0.91 | -0.90 | -0.91 |
| | F-VAE | 0.91 | 0.83 | 0.91 | 0.75 | -0.17 | -0.26 | -0.31 | -0.17 | -0.71 | -0.81 |
| **SmallNorb** | A-VAE | 0.14 | 0.09 | 0.68 | 0.56 | -0.26 | -0.04 | -0.37 | 0.28 | 0.21 | 0.77 |
| | $\beta$-VAE | 0.93 | 0.97 | 0.97 | 1.00 | 0.97 | 0.92 | 0.96 | 0.93 | 0.96 | 0.97 |
| | $\beta$-TCVAE | 0.78 | 0.94 | 0.92 | 0.99 | 0.86 | 0.97 | 0.53 | 0.76 | 0.97 | 0.98 |
| | F-VAE | 0.13 | 0.95 | 0.18 | 0.79 | -0.14 | 0.89 | -0.51 | 0.73 | 0.40 | -0.55 |

geared towards accommodating non-linear behaviours within linear subspaces tied to generative factors, fills a gap in existing literature, as no prior method aptly addressed this perspective. The proposed Latent Orthogonal Analysis (LOA) method efficiently identifies the associated subspaces and their importance. Leveraging LOA, we formulated Importance-Weighted Orthogonality (IWO), a novel metric offering unique insights into the subspace orthogonality. Throughout the experiments, our implementation emerged as a robust mechanism for assessing orthogonality, exhibiting resilience across varying latent shapes, non-linear encoding functions, and degrees of orthogonality.

**Implications** Disentanglement has long been credited for fostering fairness, interpretability, and explainability in generated representations. However, as elucidated by Locatello et al. (2019), the utility of disentangled representations invariably hinges on at least partial access to generative factors. With such access, an orthogonal subspace could be rendered as useful as a disentangled one. Through orthonormal projection, any orthogonal representation discovered can be aligned with the canonical basis, essentially achieving a state of good disentanglement.

**Comparison with Previous Work** Our IWO metric showcased superior efficacy in correlating with downstream task performance on the several datasets, where established metrics like DCI-D and MIG failed, a finding aligned with the observations of Locatello et al. (2019). We argue that sole emphasis on the explicitness of a representation may be insufficient for the evaluation of learned representations. This notion echoes the initial premise presented in the introduction. A latent space that facilitates the modulation of the smile of any person, by having distilled the independent concept of smiling is arguably different than a space that merely enables binary classification of the smiling label from encoded data points.

**Limitations** Our work is introductory. There remains a vast expanse to explore to fully understand the notion of orthogonality over disentanglement. Moreover, the real-world applicability of our metrics outside of the examined datasets and synthetic environments is yet to be fully determined.

**Conclusion** In conclusion, our work lays the groundwork for a fresh perspective on evaluating generative models. We hope the IWO metric may help identify models that craft useful orthogonal subspaces, which might have been overlooked under the prevailing disentanglement paradigm. Our hope is that IWO extends its applicability across a broader spectrum of scenarios compared to traditional disentanglement measures.

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

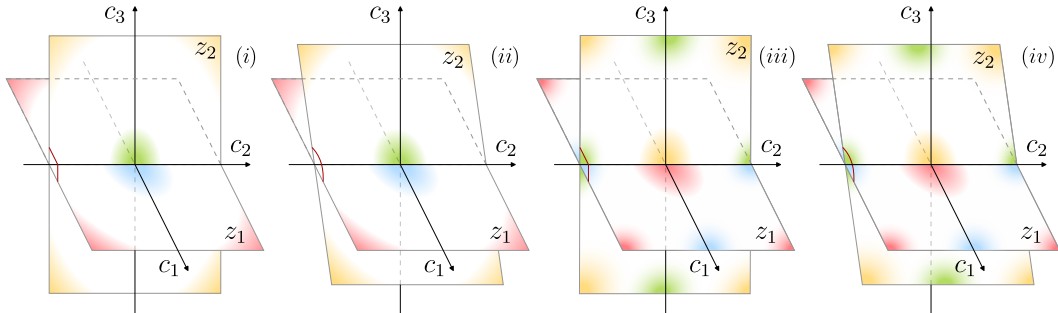

Figure 4: Four configurations of a 3-dimensional latent space. The planes represent the latent subspace where generative factors $z_1$, $z_2$ lie. The colour mapping on each subspace represents the relationship between the generative factor and the latent components (*e.g.*, blue indicating large values for $z_1$, red indicating low ones). Cases (i) and (ii) are characterized by a good explicitness score as both subspaces encode $z_1$ and $z_2$ as simple quadratic functions, contrary to cases (iii) and (iv) where the relationship is trigonometric and more complex to recover. In contrast, cases (i) and (iii) are characterized by a better IWO score compared to (ii) and (iv). Indeed, in configurations (i) and (iii), there are dimensions within each generative factor's subspace that are orthogonal to one another. Consequently, any variation along any such dimension will leave the other factor unchanged.

## A EXPLICITNESS VS IWO

In Figure 4, we depict four situations of a 3-dimensional latent space where two generative factors $z_1$, $z_2$ lie in a separate 2-dimensional plane each, and the relationship between each generative factor and its corresponding latent subspace is non-linear, as hinted by the colouring. In particular, $z_j^{(n)} = f^{(n)}(\boldsymbol{c}') = f^{(n)}(\boldsymbol{P}_j^{(n)}\boldsymbol{c})$, for $j \in \{1, 2\}$ and $n \in \{i, ii, iii, iv\}$, with $\boldsymbol{P}_j^{(n)} \in \mathbb{R}^{2\times3}$ being a projection matrix. For cases (i) and (iii), the projections span orthogonal planes, contrarily to cases (ii) & (iv) where the planes have a different inclination. Case (i) & (ii) are characterized by a quadratic relationship, *i.e.*, $f^{(n)} = A(c_1')^2 + B(c_2')^2$ (with $A$, $B$ being parameters), while case (iii) and (iv) encode a trigonometric relationship, *i.e.*, $f^{(n)} = \cos(2\pi\lambda c_1') + \cos(2\pi\lambda c_2')$ (with $\lambda$ being a parameter).

Explicitness evaluates the capacity required by a model to regress the generative factors $z_1$, $z_2$, starting from the representation $\boldsymbol{c}$. Given that the effect of the linear transformation is present in all four situations, the differences are determined only by the non-linearity $f^{(n)}$. Therefore the metric is able to discriminate cases (i) & (ii) from (iii) & (iv). Instead, IWO quantifies the orthogonality of the planes, regardless of the non-linearities in the generative factors, so it discriminates the orthogonal cases (i) & (iii) from the non-orthogonal ones (ii) & (iv). Finally, note that the DCI-Disentanglement metric would penalize all four configurations as they are not disentangled.

## B EXPERIMENTAL DETAILS

In this section, we provide a detailed description of the correlation analysis of our orthogonality metric with downstream tasks on three different datasets (Dsprites, Cars3D, SmallNorb) and four different models (Annealed VAE, $\beta$-VAE, $\beta$-TCVAE, Factor-VAE) . For each model, learned representations for six different regularization strengths are considered (ten different random seeds for each reg. strength). All these representations are retrieved from *disentanglement lib*[2]. For the ES metric, we utilized the official codebase provided by the authors of DCI-ES [3]. Each dataset we investigate has independent generative factors associated with it.

---

[2]https://github.com/google-research/disentanglement_lib/tree/master
[3]https://github.com/andreinicolicioiu/DCI-ES

## B.1 Datasets

**dSprites Dataset**   The DSprites dataset is a collection of 2D shape images procedurally generated from six independent latent factors. These factors are: color (white), shape (square, ellipse, heart), scale, rotation, and x and y positions of a sprite. Each possible combination of these latents is present exactly once, resulting in a total of 737280 unique images.

**Cars3D Dataset**   The Cars3D dataset is generated from 3D computer-aided design (CAD) models of cars. It consists of color renderings of 183 car models from 24 rotation angles, each offset by 15 degrees, and from 4 different camera elevations. The images are rendered at a resolution of $64 \times 64$.

**smallNORB Dataset**   The smallNORB dataset is designed for 3D object recognition from shape, featuring images of 50 toys categorized into five types: four-legged animals, human figures, airplanes, trucks, and cars. The images were captured under six lighting conditions, nine elevations, and 18 different angles. The dataset is split into a training set with five instances of each category and a test set with the remaining five instances.

## B.2 Standard Down Stream Tasks

The downstream tasks provided by *disentnglement lib* and considered for the correlation analysis in Section 4.2 are multi-class logistic regression (LR) and gradient boosted trees (GBT). Both concerned with simply regressing the generative factors.

## B.3 Modulation Task

In order to investigate the latent space structure, we develop a new task which we call modulation task. The task is defined as follows: Given a latent representation, a model is tasked to change a single generative factor of a representation, without changing any other. We implement this task on learned representations of the DSprites dataset for both Annealed VAE and $\beta$-TCVAE for all five factors. Table 4 holds the correlation results between IWO, IWR, DCI-D, DCI-C, and the Modulation Task Performance respectively. We see that all metrics exhibit somewhat strong correlations with this down-stream task, while IWR and IWO slightly outperform the other metrics.

## B.4 IWO on limited data

To asses the impact of smaller samples sizes on IWO and IWR's correlation with down-stream task performance, we repeat the experiments detailed in Section 4.2 for the Small Norb dataset, but with only 50% and 10% of the data. The results are illustrated in Table 3. We observe that IWO and IWR still exhibit a higher correlation with down stream task performance than other metrics, even if only 10% of the data is used for their evaluation.

## B.5 IWO Training

Given a learned representation of a dataset, we consider each generative factor independently, allocating separate LNNs respectively. On top of the LNNs, we have NN-heads, which regress the generative factors from the intermediate projections. The NN-heads are also independent from one-another and do not share any weights.

### B.5.1 Implementation Details

We use the PyTorch Lightning framework[4] for the implementation of the models required to discern IWO and IWR.

In particular we use PyTorch Lightning implementations of Linear layers and Batch-Normalization Layers. Whereas the setup of the LNNs is equal for all models and datasets, the NN-heads vary

---

[4]https://github.com/Lightning-AI/lightning

Table 3: Correlation coefficients between IWO, IWR, DCI-D, DCI-C, MIG and the downstream task of recovering the generative factors with Logistic Regression (T1) and Boosted Trees (T2). Examined Latent representations of small Norb dataset as learned by models: Annealed VAE (A-VAE), $\beta$-VAE, $\beta$-TCVAE and Factor-VAE (F-VAE) on 100%, 50% and 10% of the dataset

|  | | 1 - $\overline{\text{IWO}}$ | | $\overline{\text{IWR}}$ | | D | | C | | MIG | |
|---|---|---|---|---|---|---|---|---|---|---|---|
|  | Model | T1 | T2 | T1 | T2 | T1 | T2 | T1 | T2 | T1 | T2 |
| 100% | A-VAE | 0.14 | 0.09 | 0.68 | 0.56 | -0.26 | -0.04 | -0.37 | 0.28 | 0.21 | 0.77 |
|  | $\beta$-VAE | 0.93 | 0.97 | 0.97 | 1.00 | 0.97 | 0.92 | 0.96 | 0.93 | 0.96 | 0.97 |
|  | $\beta$-TCVAE | 0.78 | 0.94 | 0.92 | 0.99 | 0.86 | 0.97 | 0.53 | 0.76 | 0.97 | 0.98 |
|  | F-VAE | 0.13 | 0.95 | 0.18 | 0.79 | -0.14 | 0.89 | -0.51 | 0.73 | 0.40 | -0.55 |
| 50% | $\beta$-VAE | 0.20 | 0.02 | 0.43 | 0.75 | - | - | - | - | - | - |
|  | A-VAE | 0.97 | 0.98 | 0.97 | 1.00 | - | - | - | - | - | - |
|  | $\beta$-TCVAE | 0.82 | 0.95 | 0.94 | 0.99 | - | - | - | - | - | - |
|  | F-VAE | 0.13 | 0.72 | -0.09 | 0.95 | - | - | - | - | - | - |
| 10% | A-VAE | -0.02 | -0.30 | 0.50 | 0.31 | - | - | - | - | - | - |
|  | $\beta$-VAE | 0.94 | 0.99 | 0.96 | 0.99 | - | - | - | - | - | - |
|  | $\beta$-TCVAE | 0.87 | 0.95 | 0.92 | 1.00 | - | - | - | - | - | - |
|  | F-VAE | 0.32 | 0.56 | 0.12 | 0.58 | - | - | - | - | - | - |

Table 4: Correlation coefficients between IWO, IWR, DCI-D, DCI-C metrics and the downstream task of modulating a factor.

| Model | $1 - \overline{\text{IWO}}$ | $\overline{\text{IWR}}$ | DCI-D | DCI-C |
|---|---|---|---|---|
| A-VAE | 0.57 | 0.61 | 0.42 | 0.40 |
| $\beta$-TCVAE | 0.81 | 0.87 | 0.80 | 0.85 |

in their complexity for different datasets and factors. As all considered models operate with a 10-dimensional latent space, each LNN has ten layers. The output of each LNN-layer is fed to the next LNN layer and also to a the corresponding NN-head.

Table 5 holds the NN-head configuration per dataset and factor. These were found using a simple grid search on one randomly selected learned representation. This is necessary as factors vary in complexity and so does the required capacity to regress them. It is worth mentioning, that the Explicitness pipieline as proposed by Eastwood et al. (2023) could actually be employed on top of the NN-heads, as such fusing both metrics.

For the initialization of the linear neural network layers and the NN-head layers we use Kaiming uniform initialization as proposed in He et al. (2015). We further use the Adam optimization scheme as proposed by Kingma & Ba (2014) with a learning rate of $5 \times 10^{-4}$ and a batchsize of 128 for all otpimizations. Data is split into a training and a test set (80 % and 20 %). During training, part of the training set is used for validation, which is in turn used as an early stopping criteria. The importance scores used for IWO should be allocated using the test set. In our experiments the difference between the importance scores computed on the training set and on the test set were small. For further details on the implementation, please refer to our official code here: [LINK to Github].

## C  COMPUTATIONAL RESOURCES ANALYSIS

This section details the computational resources utilized for evaluating IWO and IWR, specifically applied to the smallnorb dataset from the disentanglement_lib using a beta-VAE framework. Each "run" includes parallel training of all independent models necessary for the metrics assessment. For the model specifications please refer to section B.5.1

Table 5: Implementation Details for neural network heads operating on LNN layers. For each factor, ten NN-heads with the specified number of hidden layers and their respective dimensions are trained in parallel.

| Dataset | Factor | Layer dimensions | Batch Norm | Factor Discrete |
|---------|--------|------------------|------------|-----------------|
| DSprites | Shape | 256, 256 | False | Yes |
| | Scale | 256, 256 | False | No |
| | Rotation | 512, 512, 512 | True | No |
| | x-position | 256, 256 | False | No |
| | y-position | 256, 256 | False | No |
| Cars3D | model | 256, 256, 256 | False | Yes |
| | rotation | 256, 256, 256 | False | No |
| | elevation | 256, 256, 256 | False | No |
| Small Norb | category | 256, 256 | True | Yes |
| | lightning condition | 256, 256 | True | No |
| | elevation | 256, 256 | True | No |
| | rotation | 256, 256 | True | No |

## C.1 EXPERIMENTAL SETUP

- **Data:** Learned representations of a beta-VAE trained on smallnorb dataset from the disentanglement_lib
- **Objective:** Assess the orthogonality of 60 learned representations using the IWO pipeline.

## C.2 RESOURCE UTILIZATION

| Resource | Usage |
|----------|-------|
| Process Memory | 400 MB |
| CPU Process Utilization | 50% |
| GPU Process Utilization | 0% |

Table 6: Average resource utilization for each run of the IWO pipeline. No GPU was used.

## C.3 RUNTIME ANALYSIS

- **Average Duration:** 7 minutes per run
- **Training Epochs:** 20 per run

## C.4 COMPUTATIONAL COST CONSIDERATIONS

- The inference of IWO on pretrained smallnorb representations shows minimal computational costs.
- In general, computational costs scale with the number of linear layers in the LNN spine and the capacity of the NN-heads.
- For large latent spaces, one should avoid step-wise dimensionality reduction; larger reductions between consecutive LNN-layers are preferred.
- The first LNN-layer size need not match the dimensionality of the representation. For large representations, a smaller first LNN-layer is recommended.
- GPU usage is beneficial for larger representations and models

