# OpenReview forum: "Beyond Disentanglement: On the Orthogonality of Learned Representations"
_ICLR.cc/2024/Conference — Submitted to ICLR 2024_

### Official Review · Reviewer_1Jck · 2023-10-19

**Soundness:** 3 good
**Presentation:** 3 good
**Contribution:** 3 good
**Rating:** 8
**Confidence:** 5

**Summary:**

The authors propose a novel metric for evaluating disentanglement of learned representations. The method consists of training a Linear Neural Network, essentially an MLP without nonlinearities with decreasing dimensionalities, for each ground truth factor. The objective function involves training (potentially non-linear) predictor heads on top of each hidden layer. Using QR decomposition on the learned NN weights together with loss estimates from each predictor, the authors estimate basis vectors for each ground-truth factor, together with their importance weightings. By computing such vectors for each g.t. factor they estimate both the subspaces in the learned latent space and compute a measure of orthogonality between subspaces for dfiferent g.t. factors.

Usefulness of the metric is evaluated on both synthetic and real data.

**Strengths:**

- IWO can be used in scenarios where a ground truth factor can be aligned with exactly one latent dimension
- the proposed metric actually correlates with downstream task performance

**Weaknesses:**

- only 2 datasets and models (as compared to e.g., Locatello et al. 2019) are compared in Section 4.3. Please consider using all the 7 datasets from *disentanglement_lib,* otherwise the choice seems a bit arbitrary
- Figure 1 is quite difficult to grasp. I understand that the concept is not trivial to present (and the caption is already lengthy), but maybe you could consider extending/rewriting the caption to make it clearer? Perhaps in a step wise manner (multiple figures). I find it crucial for conveying the idea of your paper. If you lack space I believe figure 3 could be compressed/removed instead

**Questions:**

The sub-optimal performance of Explicitness for the perfectly disentangled case could stem from overfitting. How do the authors handle this problem with their metric? What were the train/test splits used for the experiments? How sensitive is the metric to smaller sample sizes?

---

> ### Author Response · Authors · 2023-11-17
>
> Thank you for your constructive feedback. We have taken your comments into serious consideration and have made revisions to our manuscript accordingly.
>
> - **W1: Expansion of Dataset Usage**
>     - **Inclusion of SmallNorb Dataset**: In response to your suggestion, we have incorporated the SmallNorb dataset into our study. This dataset offers a significant variance from Dsprites and Cars3D, providing a more comprehensive evaluation of our metric.
>    - **Planned Inclusion of DSprites Variants**: For the camera-ready version of our paper, we plan to introduce three further variations of the DSprites dataset available from disentanglement_lib. This addition will further broaden the scope of our evaluation and align with the diversity presented in Locatello et al. 2019. Note that Shapes3D is unfortunately not readily available from disentanglement_lib.
>
> - **W2: Clarity of Figure 1**
>     - **Revision of Figure 1**: Following the feedback from multiple reviewers, we have replaced the original Figure 1 with a more straightforward representation in the main text, moving the original figure to the appendix. This change aims to simplify the initial understanding of our concept.
>     - **Enhanced Caption and Explanation**: As you suggested, we have elaborated on the figure's caption and provided a more detailed step-wise explanation.
>
> **Responses to Specific Questions:**
>
> - **Q1 (Handling of Overfitting)**: To mitigate overfitting, we have employed a validation set, separate from our test set, for early stopping during model training. We have added this and further training details in *Appendix B: Experimental Details*
> - **Q2 (Train/Test Splits)**: For our experiments, we reserved 15% of each dataset for testing purposes. We then used the test set for calculating the importances for the IWO metric, ensuring that our evaluations are based on unseen data. Note that the subspace basis are part of the model and learned during training.
> - **Q3 (Metric Sensitivity to Sample Sizes)**: Recognizing the importance of this concern, we have added a dedicated experiment in *Appendix B.4: IWO on Limited Data*. We investigates the sensitivity of our metric to smaller sample sizes, providing valuable insights into its robustness and reliability in varied data conditions.
>
> We appreciate the opportunity to improve our paper based on your feedback. We hope these revisions address your concerns comprehensively.

---

> > ### Comment · Reviewer_1Jck · 2023-11-17
> >
> > Thank you for updating the manuscript and the clarifications.
> > I have two more comments:
> >
> > **W1: Expansion of Dataset Usage:**
> > Shapes3D is available in disentanglement_lib, you just need to download the dataset from https://github.com/google-deepmind/3d-shapes, but the code to use it is available in the library: https://github.com/google-research/disentanglement_lib/blob/master/disentanglement_lib/data/ground_truth/shapes3d.py
> >
> > **Q3 (Metric Sensitivity to Sample Sizes)**: is there a reason why experiments on smaller datasets are missing for the baseline methods? will you include these in the camera-ready version?

---

> > > ### Author Response · Authors · 2023-11-17
> > >
> > > Thank you for your prompt response.
> > >
> > > - **W1: Expansion of Dataset Usage**: We apologize for any confusion caused by our previous comment. To clarify, in contrast to the other datasets, for Shapes3D no readily trained models are available for direct download. However, as you have pointed out, it is possible to train the models using the provided code. Because for all the other datasets pre-trained models are readily available for download, we prioritized them over Shapes3D. This enables resource efficient reproducibility of our results. However, recognizing the value of including Shapes3D, we commit to incorporating this dataset in the camera-ready version of our paper.
> > >
> > > - **Q3 (Metric Sensitivity to Sample Sizes)**: These values are still missing because they are not readily available for download from disentanglement_lib. We are however actively working on procuring these results and will include them in the camera ready version of the paper.

---

> > > > ### Comment · Reviewer_1Jck · 2023-11-17
> > > >
> > > > Thank you again for the clarification. I will update the score accordingly.

---

> > > > > ### Author Response · Authors · 2023-11-17
> > > > >
> > > > > Thank you for acknowledging our clarifications and for your willingness to update the score. We greatly appreciate your thorough review and valuable feedback throughout this process.

---

### Official Review · Reviewer_ccpN · 2023-10-30

**Soundness:** 2 fair
**Presentation:** 3 good
**Contribution:** 2 fair
**Rating:** 5
**Confidence:** 3

**Summary:**

In this paper, the authors first propose Latent Orthogonal Analysis(LOA), a method that can identify latent subspaces for different factors of variation from data. To estimate the mutual orthogonality between subspaces learned with LOA, they then propose importance-weighted orthogonality (IWO), a metric that can do the measurement on disentanglement by investigating the magnitude of the projections from different subspaces onto each other. This is achieved by multiplying the basis matric of one subspace with a diagonal matrix that defines the importance of each dimension w.r.t. the other subspace.

They empirically evaluate IWO on multiple datasets that are commonly used in disentangled representation learning, and they show that their metric that can outperform prior metrics such as MIG or DCI-D.

**Strengths:**

This paper is well-structured and clearly written. It is easy to understand what problem they try to tackle in this paper. Even though the metric study on disentangled representation learning is not a completely new field, I believe it is still worth thinking of how we evaluate the orthogonality between different subspaces that encode different factors of variation.
In their methodology, the authors provide detailed and sound math derivation on their LOA and IWO approach.

**Weaknesses:**

My main concern is about the insufficiency of evaluation. Give that $\beta$-TCVAE was a few years ago and there have been a large number of variants of VAEs that do disentangled representations, I would hope that the authors can implement a few more models for comparison. In addition, there are also very commonly used datasets that were not considered here, e.g. CelebA, Shape3D, Clevr, etc. I would like to see results on these more complex data.

**Questions:**

1. I wonder why the $\Delta$ L can be used to measure the importance. Could you justify it in more detail?
2. Is the reason that you choose to only apply linear projection using $W_{1:L}$ is technical difficulty or indeed conceptual purpose?

---

> ### Author Response · Authors · 2023-11-17
>
> Thank you for your constructive feedback. We have taken your comments into serious consideration and have made revisions to our manuscript accordingly.
>
> - **Expanded Model and Dataset Evaluation:**
> Addition of SmallNORB and Further Models: In direct response to the reviews, we have expanded our evaluation to include the SmallNORB dataset. This addition, along with the inclusion of two more models - Factor VAE and $\beta$-VAE - enriches our analysis and addresses the diversity of scenarios you highlighted. These new elements complement our existing use of Dsprites, Cars3D, Annealed VAE, and $\beta$-TCVAE. In the camera-ready version of the paper, we plan to introduce three further variations of the DSprites dataset provided by disentanglement_lib.
> - **Rationale for Dataset and Model Selection:**
>     - **Consistency with** ***disentanglement_lib***: Our decision to use the Dsprites, Cars3D, SmallNORB datasets, and the selected models is grounded in their presence within *disentanglement_lib* https://github.com/google-research/disentanglement_lib Locatello et al., 2019 [1]. This library is a benchmark in disentanglement research and offers unparalleled reproducibility and comparability. Utilizing these datasets and models minimizes biases that could arise from custom implementations or training procedures, thus providing a clear and objective evaluation of our proposed metric's efficacy.
>     - **Advantages of Established Benchmarks:**  Employing resources from *disentanglement_lib* ensures that our study aligns with prevailing research standards, enhancing the reliability and validity of our findings. This approach also allows our results to be directly comparable with a broad spectrum of existing studies, which is crucial for contextualizing and validating our contributions.
>
> **Responses to Questions:**
>
> - **Q1 (Justification of $\Delta \mathcal{L}$ as an Importance Measure)**: $\Delta \mathcal{L}_l$ measures how much the expected loss of regressing a generative factor increases when regressing from a subspace with dimension $l$ vs. $l-1$. It is therefore directly linked to the dimension which lies in the null-space of the projection from $l$ to $l-1$. As such, we use $\Delta \mathcal{L}_l$ to allocate the importance which that particular dimension (which is lost in the projection) plays for the generative factor in question.
>  In light of your feedback, we have refined the explanation in our manuscript. Additionally we have adjusted the importance-weighting mechanism in IWO to better reflect the variation in overall model informativeness, following the insights from Eastwood et al., 2018 [1].
>
> - **Q2 (Use of Linear Projections)**: Our choice to use linear projections was driven by both conceptual and technical considerations. Conceptually, linear projections are a natural fit for assessing orthogonality in latent spaces. Technically, their efficient implementation via Linear Neural Networks (LNNs) further motivated our choice.
>
> We appreciate the opportunity to improve our paper based on your feedback. We hope these revisions address your concerns comprehensively.
>
> [1] Cian Eastwood and Christopher K. I. Williams. A framework for the quantitative evaluation of dis- entangled representations. In 6th International Conference on Learning Representations, ICLR 2018, Vancouver, BC, Canada, April 30 - May 3, 2018, Conference Track Proceedings. OpenRe- view.net, 2018.

---

> ### Author Response · Authors · 2023-11-21
>
> Dear Reviewer,
>
> As the discussion period nears its conclusion, we wish to ensure that all your concerns have been adequately addressed. We would greatly appreciate any additional feedback you may have. Our goal is to finalize the rebuttal with the confidence that our manuscript reflects the valuable feedback provided by the review committee.
>
> Thank you for your time and consideration,
>
> the Authors

---

### Official Review · Reviewer_J4X7 · 2023-10-31

**Soundness:** 3 good
**Presentation:** 2 fair
**Contribution:** 2 fair
**Rating:** 3
**Confidence:** 4

**Summary:**

A new assessment scheme is introduced to measure disentanglement in latent space in terms of orthogonality between subspaces. The assessment builds on an decomposition methodology which projects the original latent encodings into incrementally smaller subspaces through linear neural models. The empirical analysis validates the proposed assessment scheme against existing disentanglement metrics on synthetic and benchmark datasets.

**Strengths:**

S1) Motivations behind the paper are solid: too strict definitions of disentanglement as projection into single orthogonal dimensions are bound to fail in realistic settings. The idea of broadening the definition to orthogonal subspaces, while not being completely novel, is developed here through an approach which is original.

S2) The technical contribution seems also solid, modulo some points which are not made entirely clear in the presentation. However, the overall methodology is convincing from the perspective of correctness and adequacy of the technical solutions.

S3) The paper is well organized and mostly of good presentation quality.

**Weaknesses:**

W1) While presentation quality is generally adequate, the paper misses to convey all the necessary details to facilitate reproduction of the method and of the study. This lack of technical detail in the main body is not compensated by the availability of appendices, supplementary materials or code. One key aspect that is unclear to me is how one is expected to identify the generative factors set $z_1,\dots, z_K$ and how such $K$ is determined in general. The method involves training a potentially large amount of regressors and little information is provided on how this is done in practice (e.g. how much should the training be pushed in terms of regression error? What are the stopping conditions? How are the linear model initialised?).

W2) The positioning with respect to the literature is on the weak end. The paper misses to discussion and cite works formalising weaker forms disentanglement [A, B, C]. In particular, it would seem relevant to discuss the relationship between the proposed approach and those building on (and measuring) linear symmetry-based disentanglement [B,D].

W3) While the experiments are generally well-designed, the evidence they provide does not seem enough to support the major claims of this paper. As long as one departs from the ideal setting, it is difficult to assess the added value of IWO over DCI and MIG. Additional experiments are needed on more challenging datasets, such as ModelNet40 and COIL-100, possibly enlarging the scope of methods to compare with by including those in [B,D]. It would also be of help to qualitatively explore the impact of the proposed methodology, e.g. by exploring the effects of manipulating the representations over the relevant subspaces “suggested” by the metrics.

W4) The proposed methodology seems very computationally involved. I am using the word “seems” as the paper lacks a comparative assessment of the cost of the method. This should be done while considering more realistically sized problems, involving latent spaces of non-trivial size.

[A] https://proceedings.neurips.cc//paper/2020/file/9a02387b02ce7de2dac4b925892f68fb-Paper.pdf

[B] https://proceedings.mlr.press/v162/tonnaer22a/tonnaer22a.pdf

[C] https://doi.org/10.1109/IJCNN55064.2022.9892093

[D] https://arxiv.org/pdf/2011.13306.pdf

**Questions:**

Q1) Can the Authors please clarify how generative factors  $z_k$ are selected for the purpose of implementing the method in general (see W1)?

Q2)  Can the Authors please discuss the relationship with linear symmetry-based metrics?

Q3)  The empirical analysis would be substantially strengthened by adding new experiments on as ModelNet40 and COIL-100, considering also computational costs?

Q4) I am a bit puzzled by the negative correlation values in the experiments: is this classical linear correlation? Because some methods seem to be highly negatively correlated (which is still somehow a form of correlation).

---

> ### Author Response · Authors · 2023-11-16
>
> Thank you for your thorough review and valuable insights. We have carefully considered each of your points and made significant revisions to our paper to address them.
>
> - **W1: Enhancing Reproducibility and Methodological Clarity**
>     - **Appendix and Implementation Details:** We have created a detailed appendix providing comprehensive information about implementation and model choices. This includes specifics on the initialization of linear layers, network training procedures, and hyper-parameter optimization strategies.
>     - **Code Repository**: To further facilitate reproducibility, we have established an anonymous code repository that contains all necessary code and documentation: https://anonymous.4open.science/r/iwo-E0C6/README.md
> - **W2: Expanding the Literature Review**
>     - **Weaker forms of Disentanglement**: We have integrated a thorough discussion on linear symmetry-based disentanglement into our related work section. This includes an analysis of the works you mentioned and others in the field, emphasizing the relationship between our approach and these methods. We have also included a segment on causal representation learning, particularly focusing on how it intersects with and differs from disentanglement methodologies, including ours.
> - **W3: Strengthening the Experimental Section**
>     - **SmallNORB Inclusion**: We agree with your point on the need for more challenging datasets. While ModelNet40 and COIL-100 are indeed valuable, we have chosen to incorporate the SmallNORB dataset beside DSprites and Cars3D. It offers comparable complexity, with diverse variations in object types, camera angles, and lighting conditions, aligning closely with the complexity of ModelNet40 and the natural imagery of COIL-100, while being part of *disentanglement_lib* https://github.com/google-research/disentanglement_lib/tree/master by Locatello et al., 2019 [1] which offers unparalleled reproducibility and comparability. Utilizing datasets and models from *disentanglement_lib* also minimizes biases that could arise from custom implementations or training procedures, thus providing a clear and objective evaluation of our proposed metric's efficacy. In the camera-ready version of the paper, we plan to introduce three further variations of the DSprites dataset provided by disentanglement_lib.
>     - **Additional Downstream Task Experiment:** We have added an experiment involving a downstream task which demonstrates the interplay between subspace structures and IWO, offering a deeper insight into the practical implications of our methodology. See *Appendix B.3: Modulation Task.*
> - **W4: Addressing Computational Costs**
>     - **Comparative Assessment**: We acknowledge the importance of understanding the computational cost of our method. We have added a section that showcases the computational requirements of our approach for 60 runs on the Small Norbs dataset. We have also added the necessary strategies to deal with high dimensional latent spaces, such as larger reductions between consecutive LNN heads and an initial LNN-layer of smaller size than the latent space.  See *Appendix C: Computational Resources Analysis*.
>
> - **Q1 (Generative Factor Identification)**: We closely align our approach with the current existing literature, using all generative factors provided by the datasets. In Appendix B.1: Datasets, all generative factors used in our analysis are listed.
> - **Q2 (Linear Symmetry-Based Metrics Relationship)**: In our expanded literature review, we discuss the relationship between our approach and linear symmetry-based metrics, delineating both the commonalities and distinctions.
> - **Q3 (Empirical Analysis Enhancement)**: We have addressed this in W3, as discussed above. The addition of the SmallNORB dataset, additional models, an analysis on reduced data (*Appendix B.4: IWO on limited data*) and the new downstream task experiment significantly bolster our empirical analysis.
> - **Q4 (Negative Correlation Values)**: The negative correlation values were also observed by  Locatello et al., 2019 [1] in their benchmark study. Indeed, both downstream task performance and disentanglement metric are taken from their disentanglement_lib repository. As to the correlations, they are calculated by averaging both metric and downstream task performance over all seeds of each hyperparameter (the hyperparameter being the model's regularization strength). The correlation therefore describes how down stream task performance and metrics correlate over the different regularization strengths. We use this numpy implementation for the computation: https://numpy.org/doc/stable/reference/generated/numpy.corrcoef.html
>
> We appreciate the opportunity to improve our paper based on your feedback. We hope these revisions address your concerns comprehensively.
>
> [1] Francesco Locatello, Stefan Bauer, Mario Lucic, Gunnar Rätsch, Sylvain Gelly, Bernhard Schölkopf, Olivier Bachem. ICML (Best Paper Award), 2019.

---

> > ### Comment · Reviewer_J4X7 · 2023-11-22
> > **Thanks for the rebuttal**
> >
> > Dear Authors, many thanks for having clarified the experimental settings and requirements of your model. I appreciate the extended literature analysis (though it would have been nice to see also the newly referenced methods compared empirically) and the improved transparency and reproducibility. Addition of SmallNORD is ok, as it is the computational analysis: the approach does not seem as computationally involved as I thought. However, it would have been good to measure the baseline computational cost also of the related models to have a means of comparison.
> > There are two aspects which still seem a limitation of the work: i) one needs to know in advance the generative factors and the method does not allow to discover those; ii) importance scores, for some reasons, should be computed on test data. The latter aspect is a bit puzzling as I was expecting that these can be computed, at least, on validation. Both issues, somewhat limit the generality and usefulness of the approach.
> > Nevertheless I will take the new information into consideration in the review internal discussion and update my score accordingly.

---

> > > ### Author Response · Authors · 2023-11-23
> > >
> > > Dear Reviewer,
> > >
> > > Thank you for your feedback and for taking the changes to our manuscript into consideration. We would like to address both limitations pointed out:
> > >
> > > **(i)** Our method is closely aligned with the setup considered by most frameworks e.g. DCI [1], MIG [2], ES [3], in that it is a metric based on known generative factors.
> > > The limitation *“one needs to know in advance the generative factors”* is a characteristic of a disentangled representation learning metric, as any attempt to estimate the generative factors without supervision is ill-posed, as pointed out by Locatello [4]. However, we were able to show that IWO and IWR are retrievable consistently from only small samples of labeled data as shown in *Appendix B.4: IWO on Limited Data*.
> > >
> > > **(ii)** Please note that we can compute IWO and IWR on the training loss, the validation loss or the test loss (this is also done by our referenced implementation). The variation in our experimental outcomes was negligible throughout these sets. However, the ability of calculating IWO and IWR on either one of these sets is, in our opinion, a strength of our method. For reference, consider the case of the DCI-D metric. This metric is commonly calculated using GradientBoosting as for example in disentanglement_lib. On that basis, impurity-based feature importances are calculated. These are not computable on test sets, which is a limitation of this approach. Because in our metric the importance is based on the regression losses, we are not limited to calculating the importance on the training set but can calculate it on any one of the training, validation or test set.
> > >
> > > **Final note:** We would like to clarify that the goal of this work is to advance the field of disentangled representation learning by expanding it through the concept of orthogonal representations. For that, precise estimation of the disentanglement score is reliant on known generative factors (full-supervision) to provide clear comparisons between different models.
> > > However, we do see future work extending our approach for weakly supervised assessment, opening doors for its application outside of foundational studies (e.g. finding generative factors through causal discovery [5, 6] or exploiting unsupervised information through the use of some regularization term in the loss of the LOA regressors).
> > > This would enable the application of our method as an unsupervised approach, further expanding the general use of our formulation and method.
> > >
> > > Thank you for your time and consideration,
> > >
> > > Kind regards,
> > >
> > > the Authors
> > >
> > > [1] https://openreview.net/pdf?id=By-7dz-AZ
> > >
> > > [2] https://arxiv.org/pdf/1802.04942.pdf
> > >
> > > [3] https://arxiv.org/pdf/2210.00364.pdf
> > >
> > > [4] https://arxiv.org/abs/1811.12359
> > >
> > > [5] https://proceedings.mlr.press/v177/faria22a.html
> > >
> > > [6] https://www.jmlr.org/papers/volume24/21-1329/21-1329.pdf

---

> ### Author Response · Authors · 2023-11-21
>
> Dear Reviewer,
>
> As the discussion period nears its conclusion, we wish to ensure that all your concerns have been adequately addressed. We would greatly appreciate any additional feedback you may have. Our goal is to finalize the rebuttal with the confidence that our manuscript reflects the valuable feedback provided by the review committee.
>
> Thank you for your time and consideration,
>
> the Authors

---

### Official Review · Reviewer_Voxk · 2023-11-01

**Soundness:** 3 good
**Presentation:** 1 poor
**Contribution:** 3 good
**Rating:** 5
**Confidence:** 3

**Summary:**

This paper considers the question of evaluating the quality of disentangled representations with respect to the orthogonality of factors. The authors propose latent orthogonal analysis used to devise a new metric called importance weighted orthogonality. The method is evaluated on several datasets and shows promising results.

**Strengths:**

The research question is strong and to the best of my knowledge this problem is still open at large, and so any advancement on this front is highly important. Another strength is the relative simplicity of the approach, involving basic neural networks and standard linear algebra operations. The results are also compelling, although somewhat basic, in my opinion.

**Weaknesses:**

The main weakness of this submission is the clarity of exposition. In particular, Sections 2 and 3 could be improved significantly. For instance, the illustration in Fig. 1 is unclear. I believe the authors could do better by considering a 2D case instead of 3D, minimizing the use of colors and angles in the figure. Further, several crucial algorithmic components are described in a minimal fashion with supporting equations,illustrations, or pseudo-code. For example, the text above Eq. (2) and the text above Eq. (4). Given that the proposed method does not seem to be overly complex, I find it disappointing that its description is somewhat vague.

Another weakness is the evaluation section. Evaluating disentangled factors is a long-standing problem in representation learning. In particular, there are established benchmarks and papers focused on this particular problem. While I am not an expert on this issue specifically, I would assume that suggesting a new metric that is arguably better than others should be motivated better and empirically justified with more than two real-world datasets and a few toy examples.

**Questions:**

See above

---

> ### Author Response · Authors · 2023-11-16
>
> Thank you for your insightful review and constructive feedback. We have made several improvements to our manuscript based on your comments.
>
> - **Figure 1 Clarity and Simplification**
> In response to your suggestion, we have revised Figure 1 to present a 2D case, which simplifies the concept and minimizes the use of colors and angles. The figure now describes the difference between disentangled and orthogonal representations, in terms of DCI-D and IWO. As for the comparison with the Explicitness, acknowledging the original importance of the figure, as highlighted by Reviewer [1Jck], we have retained it in the appendix with an expanded caption for a more in-depth understanding.
>
> - **Enhanced Algorithmic Description**
> We have thoroughly revised Sections 2 and 3 to provide a clearer exposition of our methodology. This includes a more logical structure in presenting our formulas, where we first elaborate on the concept and purpose of e.g. IWO, before delving into its algebraic representation. This approach ensures a more intuitive understanding of our method and its theoretical underpinnings.
>
> - **Comprehensive Evaluation Section**
> To address your concerns regarding the evaluation of disentangled factors, we have expanded our experimental section to align closely with the benchmark established by Locatello et al., 2019 [1], recognized as a gold standard in disentanglement research. We now include analyses on dSprites, SmallNORB, and Cars3D datasets. In the camera-ready version of the paper, we plan to introduce three further variations of the dSprites dataset. This expansion not only adheres to established benchmarks but also showcases the versatility and robustness of our proposed metric across a broader spectrum of scenarios.
>
> We hope these revisions address your concerns comprehensively and enhance the overall quality and clarity of our work.
>
> [1] Francesco Locatello, Stefan Bauer, Mario Lucic, Gunnar Rätsch, Sylvain Gelly, Bernhard Schölkopf, Olivier Bachem. ICML (Best Paper Award), 2019.

---

> ### Author Response · Authors · 2023-11-21
>
> Dear Reviewer,
>
> As the discussion period nears its conclusion, we wish to ensure that all your concerns have been adequately addressed. We would greatly appreciate any additional feedback you may have. Our goal is to finalize the rebuttal with the confidence that our manuscript reflects the valuable feedback provided by the review committee.
>
> Thank you for your time and consideration,
>
> the Authors

---

> > ### Comment · Reviewer_Voxk · 2023-11-22
> >
> > Dear Authors,
> >
> > I would like to thank you for your response. I also went over the revised version. I still feel that my main concerns (poor exposition and basic evaluation) are not fully addressed, unfortunately. Nevertheless, I will take into account your rebuttal and revised manuscript during the reviewers' discussion.

---

> > > ### Author Response · Authors · 2023-11-23
> > >
> > > Dear Reviewer,
> > >
> > > We appreciate the consideration of our rebuttal and revised manuscript. Taking your feedback into account we would like to further address your concerns.
> > >
> > > Regarding the **exposition**, we have edited the methodology section, to further improve the clarity. We have ensured that our method is self-sufficient, with Figures 2 and 3 serving as supplementary aids rather than necessities. In particular, we have:
> > > - addressed any ambiguity that our notation may have caused
> > > - better clarified the aim of LOA
> > > - motivated the design choices of our method
> > > - specifically added a formula for the retrieval of basis vectors from the linear neural network
> > > - enhanced the motivation behind our importance scores
> > >
> > >
> > > Regarding the **concerns on the evaluation of our work**, we wish to point out that, in addition to our comprehensive synthetic experiments, we have evaluated our approach using three benchmark datasets (Dsprites, Cars3D, SmallNorb). These evaluations were conducted with four benchmark models ($\beta$-VAE, Annealed VAE, $\beta$-TCVAE, Factor VAE) that are prevalently utilized in disentanglement research. Our methodology was compared against four widely used metrics (DCI-D, DCI-C, MIG, DCI-ES). Moreover, we have expanded our analysis to include scenarios with limited data availability (see Appendix B.4). In response to the insightful feedback from reviewer 1Jck, we are also incorporating the Shapes3D dataset and three variations of the DSprites dataset into our evaluation process for the camera ready version of our paper.
> > >
> > > Thank you for your time, we hope these revisions address your concerns and enhance the overall quality and clarity of our work.
> > >
> > > Kind regards,
> > >
> > > the Authors

---

### Author Response · Authors · 2023-11-16

We are grateful for the insightful and constructive feedback provided by the reviewers. We are pleased to read that our research question is *strong* [VoxK], our technical contribution and math derivation are *solid* and *sound* [J4X7, ccpN] and that the paper is *well organized*, *clearly written* and of *good presentation quality* [J4X7, ccpN].

Furthermore, the constructive suggestions have been instrumental in refining our manuscript and amplifying its contributions to the field. We have carefully considered and addressed each point raised, ensuring that our research not only remains robust but also becomes more accessible.

- **Experiments**:
We have enriched our experimental setup with the SmallNorb dataset and additional models as suggested, enhancing the comprehensiveness and robustness of our findings. Our revised Table 2, now with improved readability through colour-coding, better demonstrates our methodology’s effectiveness.

- **Figure 1 Redesign**:
We have modified Figure 1 for greater clarity, aligning with the reviewers recommendations. The new 2D representation succinctly conveys the intended concept, making it more digestible for the readers.

- **Expanded Related Work**:
Our expanded section on weak disentanglement and symmetry-based learning provides a richer context for our work, drawing clearer distinctions and similarities with existing literature.

- **Methodological Refinements**:
Responding to reviewer [ccpN], we’ve refined the importance weighing in IWO and IWR, we have also adapted IWR's calculation, enhancing the metrics’ resilience and alignment with established standards.

- **Enhanced Readability**:
The method section has been revised for improved clarity and readability, ensuring that our approach is easily comprehensible and accessible.

---

### Meta-Review · Area_Chair_gUP7 · 2023-12-13

**Metareview:**

This paper proposes an alternative metric to evaluate disentanglement and demonstrate enhanced correlation with downstream task performance.

This paper was quite borderline. One reviewer was positive, while the other three recommended rejection. The more negative reviewers appreciated the changes made by the authors to the paper, but during the discussion, they thought that the paper was still not ready for publication at ICLR. In particular, they mentioned that while Section 3's clarity had improved, it was still "a hard read for a relatively straightforward algorithm". Second, while the authors added another dataset to their evaluation, there are still several important missing benchmarks, and the reviewer wanted to properly review the additional experiments that the authors promised before potentially accepting the paper. As the ideas were deemed on the incremental side, a more solid experimental section was required.

Without stronger support for this paper, I decided to reject it. I encourage the authors to take the detailed feedback from the reviewers to improve their manuscript and re-submit at the next machine learning conference.

**Justification For Why Not Higher Score:**

There was not sufficient support for this paper among the reviewers. See the meta-review for concerns to address in the revision.

**Justification For Why Not Lower Score:**

N/A

---

### Decision · Program_Chairs · 2024-01-16

Reject